# Technical note: Common ambiguities in plant hydraulics

Yujie Wang[1] and Christian Frankenberg[1,2]

[1]Division of Geological and Planetary Sciences, California Institute of Technology, Pasadena, California 91125, USA
[2]Jet Propulsion Laboratory, California Institute of Technology, Pasadena, California 91109, USA

**Correspondence:** Yujie Wang (wyujie@caltech.edu)

**Abstract.** Plant hydraulics gains increasing interest in plant eco-physiology and vegetation modeling. However, the hydraulic properties and profiles are often improperly represented thus leading to biased results and simulations, e.g., the neglection of gravitational pressure drop results in overestimated water flux. We highlight the commonly seen ambiguities and/or misunderstandings in plant hydraulics, including (1) distinction between water potential and pressure, (2) differences among hydraulic conductance and conductivity, (3) xylem vulnerability curve formulations, (4) model complexity, (5) stomatal model representations, (6) bias from analytic estimations, (7) whole plant vulnerability, and (8) neglected temperature dependencies. We recommend careful thinking before using or modifying existing definitions, methods, and models.

## 1 Introduction

Plant hydraulics gains increasing interest in understanding plants' responses and acclimation/adaptation to the environment (Santiago et al., 2004; McDowell et al., 2008; McDowell, 2011; Meinzer et al., 2010; Allen et al., 2010; Anderegg et al., 2012, 2016; Gleason et al., 2016; Wang et al., 2021a; Liu et al., 2021) and modeling canopy carbon and water fluxes within vegetation and land surface models (Buckley and Mott, 2013; Manzoni et al., 2013; McDowell et al., 2013; Sperry et al., 2017; Kennedy et al., 2019; Liu et al., 2020; Wang et al., 2020, 2021b; Sabot et al., 2022). However, xylem hydraulic properties and flow pressure profile are often improperly represented, due to the ambiguities and misunderstandings of various plant hydraulic parameters, though the plant hydraulic models used in topical research are already dramatically simplified compared to a complicated hydraulic architecture (Tyree and Ewers, 1991). For instance, distinctions between (a) water potential and pressure, (b) hydraulic conductance and conductivity, and (c) division and derivative are often not recognized. Further, the pursuit of simplicity, analytical solution, and novelty consequently results in modifications of known and well tested functional forms. However, while researchers should be encouraged to try "new" approaches, it is important to keep in mind whether these changes or new methods (a) are correct and (b) need to be tested before moving forward. Any research violating the two principles would be unwarranted, not matter how "reasonable" they appear to be.

For example, regarding the modeling of plant hydraulics, since Wolf et al. (2016) and Sperry et al. (2017) advanced the stomatal optimization theory (Cowan and Farquhar, 1977) by quantifying hydraulic risk under a general gain-risk optimization framework, an increasing number of new models or variants have been developed (e.g., Anderegg et al., 2018; Dewar et al., 2018; Eller et al., 2018; Wang et al., 2020, 2021a; Chen et al., 2022); and many plant hydraulics-based models show predictive skills comparable to statistical methods (Anderegg et al., 2018; Venturas et al., 2018; Eller et al., 2020; Wang et al., 2020;

Sabot et al., 2022). Nevertheless, these tested models are not always replicated correctly as researchers tend to mutate the formulations and sometimes hypotheses, such as the neglect of the rhizosphere component that plays an important role in drought stress conditions (Sperry et al., 1998; Sperry and Love, 2015; Sperry et al., 2016; Wang et al., 2020). Although the

modifications often resemble tested models, they are often used without being thoroughly tested. Reasons behind the lack of model testing include (a) there is not yet a well established method or database to conveniently benchmark the new model variants, (b) research that focuses on varying the formulations and testing the variants is not encouraged by reviewers due to lack of novelty, and (c) one may not recognize the changes or differences that have been made. Here, we list some common ambiguities and misunderstandings in plant hydraulics, and recommend careful thinking before using or modifying existing

definitions, methods, and models.

## 2   Water potential and pressure

Water movement in xylem conduits includes mass flow through xylem conduits and diffusion between xylem conduits and capacitance tissues. Water mass flow (from site 1 to 2) in xylem is driven by the net force at the target plane per area (driving pressure, DP; see Table 1 for the list of symbols), which is $DP = P_{x1} - P_{x2} + \rho g h_1 - \rho g h_2$ as in Figure 1a. Dissolved ions only

play a role in liquid water density but do not contribute to the driving pressure for long-distance transport in the xylem, because the distance involved is too long for diffusion and thus chemical potential to be an important contributor. Water diffusion across the cell membrane (from xylem conduit to the cell) is driven by the potential difference, which is $P_x - P_c + \Psi_{sx} - \Psi_{sc}$ as in Figure 1a; and the dissolved ions play a role through the osmotic potential.

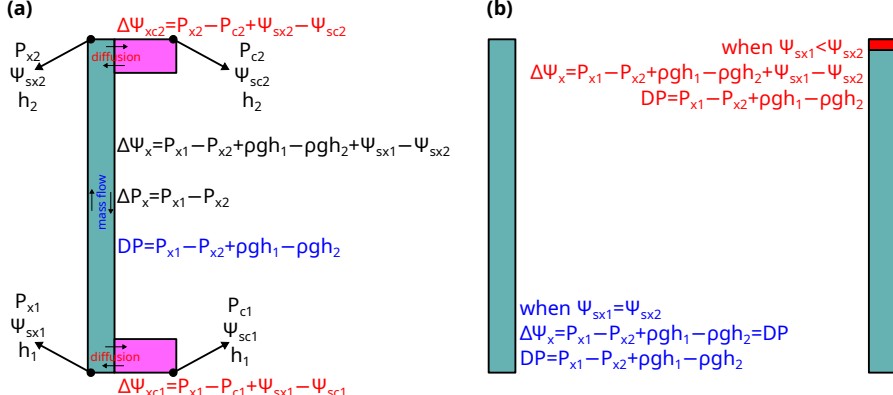

**Figure 1.** Diagram for the difference between potential and pressure in plant hydraulics. **(a)** Difference among pressure difference ($\Delta P$), potential difference ($\Delta \Psi$), and mass flow driving pressure (DP). $\Psi$ is the sum of water pressure ($P_x$ for xylem and $P_c$ for cell), osmotic potential ($\Psi_{sx}$ for xylem and $\Psi_{sx}$ for cell), and gravitational potential related to height ($\rho g h$). **(b)** Example when $\Delta \Psi$ differs from $\Delta P$. In scenario 1 (left) where $\Psi_{sx2} = \Psi_{sx1}$ (labeled in blue), the $\Delta \Psi = DP$. In scenario 2 (right) where $\Psi_{sx2} < \Psi_{sx1}$ (labeled in red), $\Delta \Psi > DP$. See Table 1 for the list of symbols.

**Table 1.** List of symbols.

| Symbol | Description | Unit |
|---|---|---|
| DP | Driving pressure ($P_{x1} - P_{x2} + \rho g h_1 - \rho g h_2$) | MPa |
| $P$ or $p$ | Water pressure | MPa |
| $P_c$ | Cell turgor pressure | MPa |
| $P_x$ | Xylem water pressure | MPa |
| $\Psi$ or $\psi$ | Water potential | MPa |
| $\Psi_s$ | Osmotic potential from dissolved solute | MPa |
| $\Psi_{sc}$ | $\Psi_s$ of living cells | MPa |
| $\Psi_{sx}$ | $\Psi_s$ of xylem sap | MPa |
| $A_L$ | Leaf area the xylem supports | $m^2$ |
| $A_S$ | Sapwood area | $m^2$ |
| $k$ | Hydraulic conductance | $mol\ MPa^{-1}\ s^{-1}$ |
| $k_{branch}$ | Hydraulic conductance of the branch | $mol\ MPa^{-1}\ s^{-1}$ |
| $k_L$ | Leaf area specific hydraulic conductance | $mol\ MPa^{-1}\ m^{-2}\ s^{-1}$ |
| $K$ | Hydraulic conductivity | $mol\ m\ MPa^{-1}\ s^{-1}$ |
| $K_L$ | Leaf area specific hydraulic conductivity | $mol\ MPa^{-1}\ m^{-1}\ s^{-1}$ |
| $K_S$ | Sapwood area specific hydraulic conductivity | $mol\ MPa^{-1}\ m^{-1}\ s^{-1}$ |
| $Q$ | Flow rate through the xylem segment | $mol\ s^{-1}$ |
| $A_{max}$ | Maximum achievable photosynthetic rate | $\mu mol\ m^{-2}\ s^{-1}$ |
| $E$ | Transpiration rate of the whole plant | $mol\ s^{-1}$ |
| $E_{crit}$ | Maximum $E$ beyond which the plant desiccate | $mol\ s^{-1}$ |
| $k_{canopy}$ | Marginal hydraulic conductance of the canopy ($dE/dP_{canopy}$) | $mol\ MPa^{-1}\ s^{-1}$ |
| $k_{canopy,ref}$ | $k_{canopy}$ when transpiration rate is 0 | $mol\ MPa^{-1}\ s^{-1}$ |
| $k_{plant}$ | Whole plant hydraulic conductance | $mol\ MPa^{-1}\ s^{-1}$ |
| $P_{canopy}$ | Water pressure at the end of leaf xylem | MPa |
| $\Psi_{soil}$ | Soil water potential | MPa |
| $\Theta$ | Risk associated with stomatal opening | $\mu mol\ m^{-2}\ s^{-1}$ |
| VC | Vulnerability curve | - |
| $a, b$ | Logistic function parameters | -, $MPa^{-1}$ |
| $B, C$ | Weibull function parameters | MPa, - |
| $k_{max}$ | Maximum hydraulic conductance | $mol\ MPa^{-1}\ s^{-1}$ |
| $k_{max,25}$ | Maximum hydraulic conductance at 25 °C | $mol\ MPa^{-1}\ s^{-1}$ |
| $m, n$ | Power function parameters | $MPa^{-n}$, - |
| $P_{50}$ | Xylem water pressure where xylem loses 50% conductance | MPa |
| $\eta, \eta_{25}$ | Viscosity of water (at 25 °C) | Pa s |
| $\gamma, \gamma_{25}$ | Surface tension of water (at 25 °C) | $N\ m^{-1}$ |

As ion concentration in xylem conduit is very low, the osmotic potential in xylem conduit ($\Psi_{sx}$) is often ignored. Therefore, water potential is imperceptibly used in place of mass flow driving pressure (i.e., DP) because of the gravity term in it. To date, many people use water potential rather than water pressure when modeling and describing mass water flow. This, though may be easier for people to understand in most scenarios, is not correct. For example, in Figure 1b, if the osmotic potential at the bottom and top are the same, the driving pressure and water potential difference are the same. However, if the osmotic potential at the top is more negative than at the bottom (for example, via adding a very thin layer of high concentration salt solution), the driving pressure will be lower than the water potential difference. Besides the fact that the values of DP and water potential difference do not always equal, the primary reason for not misuse them is that water potential describes the tendency for water to move between adjacent phases (where water molecules will diffuse), whereas pressure is more relevant to bulk water movement. Thus, using potential difference for water mass flow is technically incorrect, and it is necessary to clarify the terminology to distinguish them.

Water potential and pressure used in plant hydraulics are both defined as a difference from a reference value: water potential (often denoted as $\Psi$ or $\psi$ in literature) is typically defined as the difference from the potential of pure water in the soil, and water pressure (often denoted as $P$ or $p$) is typically defined as the difference from the environmental air pressure. Water pressure difference between xylem water and surrounding air is responsible for air-seeded conduit cavitation (Sperry and Tyree, 1988; Tyree and Sperry, 1989), which occurs when the pressure difference exceeds the capillary pressure at the air-water interface. Thus, using water potential to describe xylem vulnerability curve should be avoided. Although the $\Delta$ values of the two are interchangeable in many scenarios (e.g., when there is no height change or external air pressure, and osmotic potential in the xylem is zero), one needs to be cautious to avoid ambiguity:

– Use pressure in xylem cavitation;

– Use pressure in water mass flow;

– Use potential in water diffusion across the cell membrane (e.g., water exchange between xylem and living cell).

We note that water transport in plants also include mass flow within the apoplastic spaces (e.g., in roots and leaves; Aloni et al. (1998)) and through plasmodesmata (e.g., between bundle sheath and phloem; Schulz (2015)), liquid water diffusion among living cells, and gaseous vapor phase diffusion among water-air interfaces (e.g., vapor diffusion within the stomatal chamber; Buckley (2015); Buckley et al. (2017)). As recommended, it is more accurate to use potential for diffusion and pressure to mass flow.

A commonly seen mistake is the use of leaf water potential to describe measurements from the pressure chamber method (Scholander et al., 1964; Boyer, 1967), which gives a decent estimate of xylem water pressure. People often refer to the measurement as leaf water potential as (a) xylem conduit water has very low solute content, (b) gravity term is often negligible compared to the very negative leaf xylem water pressure, and (c) if the water has reached equilibrium internally prior to the pressure chamber measurement, xylem water potential should equal that in the mesophyll. However, it is always more accurate to treat it as an equivalent pressure or a balance pressure (at the end of xylem). Similar logic applies to xylem water potential

and xylem water pressure, and so does the thermocouple psychrometers method (Boyer and Knipling, 1965; Boyer, 1968). It is recommended to refer to the measurement as leaf/xylem water pressure or balance pressure in the future, rather than leaf/xylem water potential that is not directly measurable.

## 3 Hydraulic conductance and conductivity

Hydraulic conductance ($k$) and conductivity ($K$) are also often confused in the literature (e.g., Kannenberg et al., 2019; Cardoso et al., 2020; Li et al., 2021). Hydraulic conductance (flow rate divided by driving pressure) is an extensive property (depends on the extent/size of the system), whereas hydraulic conductivity is an intensive property that is supposed to represent different xylem anatomy. The most widely used definitions for conductance and conductivity are: (a) hydraulic conductance (namely $k$) is the ratio between flow rate through the segment ($Q$) and driving pressure ($\Delta P - \rho g \Delta h$) (an extensive parameter depending on segment length and cross-section area), (b) hydraulic conductivity (namely $K$) is the ratio between flow rate and driving pressure gradient (an extensive parameter depending on segment cross-section area), (c) sapwood area specific hydraulic conductivity ($K_S$) is the ratio between hydraulic conductivity and xylem sapwood area ($A_S$), and (d) leaf area specific hydraulic conductivity ($K_L$) is the ratio between hydraulic conductivity and leaf area the xylem supports ($A_L$):

$$k = \frac{Q}{\Delta P - \rho g \Delta h},\tag{1}$$

$$K = k \cdot L,\tag{2}$$

$$K_S = \frac{K}{A_S},\tag{3}$$

$$K_L = \frac{K}{A_L}.\tag{4}$$

Note that only $K_S$ and $K_L$ are per unit conducting area, and thus can be treated as "intrinsic" properties for comparison purpose: $K_S$ for sapwood water permeability and $K_L$ for leaf water supply capability. However, $K_L$ may not best describe leaf water supply capability. For example, if two branches have the same $K_S$, leaf area, and sapwood area, but only differ in their length, the computed $K$ and $K_L$ would be the same for the two branches even though the actual leaf water supply capabilities differ. In comparison, conductance of the entire branch divided by leaf area of the branch, i.e., leaf area specific hydraulic conductance ($k_L$) as inspired by leaf area specific whole plant hydraulic conductance, would be a better measure for leaf water supply. The $k_L$ can be estimated using

$$k_L = \frac{k_{branch}}{A_L},\tag{5}$$

where $k_{branch}$ is hydraulic conductance of the entire branch (not a stem segment).

## 4 Xylem vulnerability curve (VC)

Various formulas have been used to represent xylem VC, and the three most common ones are Weibull cumulative probability function (equation 6) (e.g., Sperry et al., 2016; Love et al., 2019), logistic function (equation 7) (e.g., Feng et al., 2018; Huber

et al., 2019), and power function (equation 8) (e.g., Eller et al., 2018; Liu et al., 2020):

$$\frac{k}{k_{\max}} = \exp\left[-\left(\frac{-P}{B}\right)^C\right] = 2^{-\left(\frac{P}{P_{50}}\right)^C}, \tag{6}$$

$$\frac{k}{k_{\max}} = 1 - \frac{1}{1 + a \cdot \exp(b \cdot P)} = 1 - \frac{1}{1 + \exp[b \cdot (P - P_{50})]}, \tag{7}$$

$$\frac{k}{k_{\max}} = \frac{1}{1 + m \cdot (-P)^n} = \frac{1}{1 + \left(\frac{P}{P_{50}}\right)^n}, \tag{8}$$

where $B$, $C$, $a$, $b$, $m$, and $n$ are vulnerability function parameters, and $P_{50}$ is the water pressure at which the tissue loses 50% of its conductance. Note that there are also more complex VC formulations based on the three, such as dual-Weibull function used in hydraulic fiber bridge (Cai et al., 2014; Pan and Tyree, 2019) and cavitation fatigue (Feng et al., 2015; Zhang et al., 2018).

We should be aware that the logistic VC function (equation 7, or formulation based on it) does not always start from 1 when $P = 0$. This problem is minor for sigmoidal VCs (s-shaped); however, the offset at $P = 0$ could introduce bias if the VC becomes more exponential (r-shaped, see Fig. 5 of Huber et al. (2019) for an example). In this case, fitting VC using equation 7 would result in overestimated $k_{\max}$ and less negative $P_{50}$. Thus, equation 7 should be rescaled to minimize the bias, and the modified formulation is

$$\frac{k}{k_{\max}} = \frac{a \cdot \exp(b \cdot P)}{1 + a \cdot \exp(b \cdot P)} \cdot \frac{1 + a}{a} = \frac{(1 + a) \cdot \exp(b \cdot P)}{1 + a \cdot \exp(b \cdot P)}, \tag{9}$$

$$P_{50} = -\frac{\log(2 + a)}{b}. \tag{10}$$

## 5 Hydraulic model complexity

Plant hydraulic models have various complexities depending on the various aims of research and difficulties in model parameterization (Tyree and Ewers, 1991; Tyree and Zimmermann, 2002). In terms of flow profiles, the models can be categorized to steady state and non-steady state models. The steady state models use a constant flow rate within roots, stem, and leaves. The non-steady state models employ a changing flow rate within or among different tissues given the water exchange between xylem and capacitance tissues. In terms of the model complexity, the models range from a single element to a xylem network (say multiple roots and multiple canopy layers). Further, hydraulic conductance of an element may change with the growth of plants; for example, the drought legacy, maximum hydraulic conductance, and VC vary with the stack of new tree rings (McCulloh and Sperry, 2005; Cai and Tyree, 2010). Although more complex models may better represent the water flow and pressure profiles within the plants, increasing difficulties in model parameterization makes these more complex models less appealing to users. However, inappropriate model selection could result in biased results, for instance, modeling plant hydraulics at steady state for plants with high water capacity and ignoring vessel tapering effect when modeling xylem growth. Thus, it is important to select plant hydraulic models with adequate complexity in topical research. See the section below for a detailed

example of how reduced model complexity (ignoring VC segmentation) may bias the modeled hydraulic risk and thus stomatal responses.

## 6 Stomatal model representation

Plant hydraulics-based stomatal models are gaining increasing interest in the vegetation and land modeling communities (e.g., Kennedy et al., 2019; Sabot et al., 2020) as they predict stomatal closure at dry environmental conditions without employing an arbitrary tuning factor (often known as the $\beta$ factor) (Powell et al., 2013). For instance, the recently developed optimality theory-based models propose that plants should balance the gain and risk associated with stomatal functioning (Wolf et al., 2016; Sperry et al., 2017). When plants open their stomata more, plants gain more photosynthetic carbon, but lose more water and have higher risk in hydraulic failure; therefore, plants are supposed to find a sweet zone to maximize the difference between the gain and risk. These optimality theory models, particularly those weigh the risk based on plant hydraulics, show comparable or better predictive skills compared to the statistical approaches (Anderegg et al., 2018; Eller et al., 2018; Venturas et al., 2018; Wang et al., 2020; Sabot et al., 2022). However, a common mistake when using plant hydraulics-based models is that one does not follow the original model formula or hypothesis. For example, the Sperry et al. (2017) model defines the risk associated with stomatal functioning ($\Theta$) as

$$\Theta = A_{\text{max}} \cdot \frac{k_{\text{canopy,ref}} - k_{\text{canopy}}(P_{\text{canopy}})}{k_{\text{canopy,ref}} - k_{\text{crit}}}, \tag{11}$$

$$k_{\text{canopy}} = \frac{\mathrm{d}E}{\mathrm{d}P_{\text{canopy}}}, \tag{12}$$

where $A_{\text{max}}$ is the maximal achievable photosynthetic rate at the given setting, $P_{\text{canopy}}$ is the water pressure at the end of leaf xylem, $k_{\text{canopy,ref}}$ is the maximum $k_{\text{canopy}}$ when transpiration rate is 0, $k_{\text{crit}}$ is the $k_{\text{canopy}}$ when transpiration rate reaches the maximum transport capacity of the xylem, and $E$ is the transpiration rate. The $k_{\text{crit}}$ by definition is 0, as a minimum incremental transpiration rate results in infinity increase in xylem pressure (the $\mathrm{d}E/\mathrm{d}P_{\text{canopy}} = 0$ in Figure 2). Thus, in the subsequent research where Sperry et al. (2017) was tested (Venturas et al., 2018; Wang et al., 2020), the model has been reformulated to

$$\Theta = A_{\text{max}} \cdot \left[ 1 - \frac{k_{\text{canopy}}(P_{\text{canopy}})}{k_{\text{canopy,ref}}} \right]. \tag{13}$$

Note here that $k_{\text{canopy}}$ is the derivative of a water supply curve at given soil water potential and canopy water pressure, and $k_{\text{canopy}}/k_{\text{canopy,ref}}$ is different from (a) relative conductance of root, stem, or leaf xylem (i.e., $k/k_{\text{ref}}$, where $k_{\text{ref}}$ is the maximum $k$ at a reference xylem pressure), and (b) relative whole-plant hydraulic conductance ($k_{\text{plant}} = E/\mathrm{DP}$). However, model descriptions from various sources may be contrasting: Sperry et al. (2017) expressed their risk in derivative form; Eller et al. (2018) expressed their risk calculations in three completely different ways including division (their equation 2.3), derivative (their equation 2.6), and point estimation (their equation 2.8); Mencuccini et al. (2019) interpreted the two models based on $k_{\text{plant}}$; Wang et al. (2020) interpreted the two models based on the derivative forms. See Fig. 2 for how the three quantities differ.

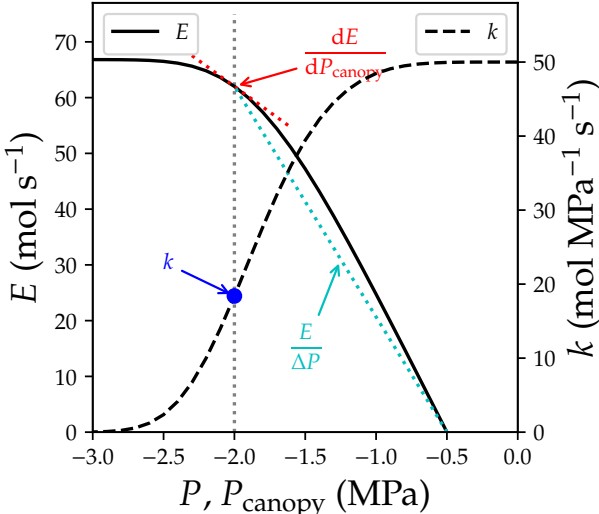

**Figure 2.** Difference between quantities used in plant hydraulics. The simulation is done for a plant with soil water potential of $-0.5$ MPa, no gravity term, and no drought legacy effect from previous xylem embolism. See Table 1 for the list of symbols.

For a xylem that does not have height change or VC segmentation, $\mathrm{d}E/\mathrm{d}P_{\text{canopy}} = k(P)$. Otherwise, using $k(P)$ to proxy $\mathrm{d}E/\mathrm{d}P_{\text{canopy}}$ could result in biases, particularly when gravity is not negligible and when tissue VCs differ dramatically (Sperry et al., 2016; McCulloh et al., 2019). Using the parameters of a real plant as an example (data from Wang et al., 2019), it is obvious that none of the root, stem, leaf, or whole-plant hydraulic conductance is a good $\mathrm{d}E/\mathrm{d}P_{\text{canopy}}$ proxy (Fig. 3). Therefore, researchers should test the models that differ from the original forms. To note, the primary reason that Sperry et al. (2017) used $\mathrm{d}E/\mathrm{d}P_{\text{canopy}}$ was to account for the VC segmentation. Using stem VC (easiest to measure; typically more resistant than roots and leaves) to proxy root and leaf VCs in stomatal models would likely result in less sensitive stomatal response to environmental stimuli such as soil moisture (Fig. 3).

## 7 Analytic solution and estimation

The pursuit of simplicity and analytic solution often leads to biased results, for example, using leaf or stem VC as a proxy for $\mathrm{d}E/\mathrm{d}P_{\text{canopy}}$ (e.g., Fig. 3) and ignoring the impact of gravity (e.g., Fig. 4). As a result, it is important to distinguish true analytic solution from analytic estimation. For example in a xylem water supply curve, when gravitational pressure drop is neglected, flow rate at a given canopy water pressure will be overestimated (Fig. 4). The more the height changes, the more $E$ is overestimated (Fig. 4). The absolute value of $E$ overestimation decreases with more negative soil water potential, whereas the relative $E$ overestimation increases with more negative soil water potential (Fig. 4). Thus, given the potentially great biases, it is recommended to verify any analytic or numerical estimations against true numerical solution before using them in research.

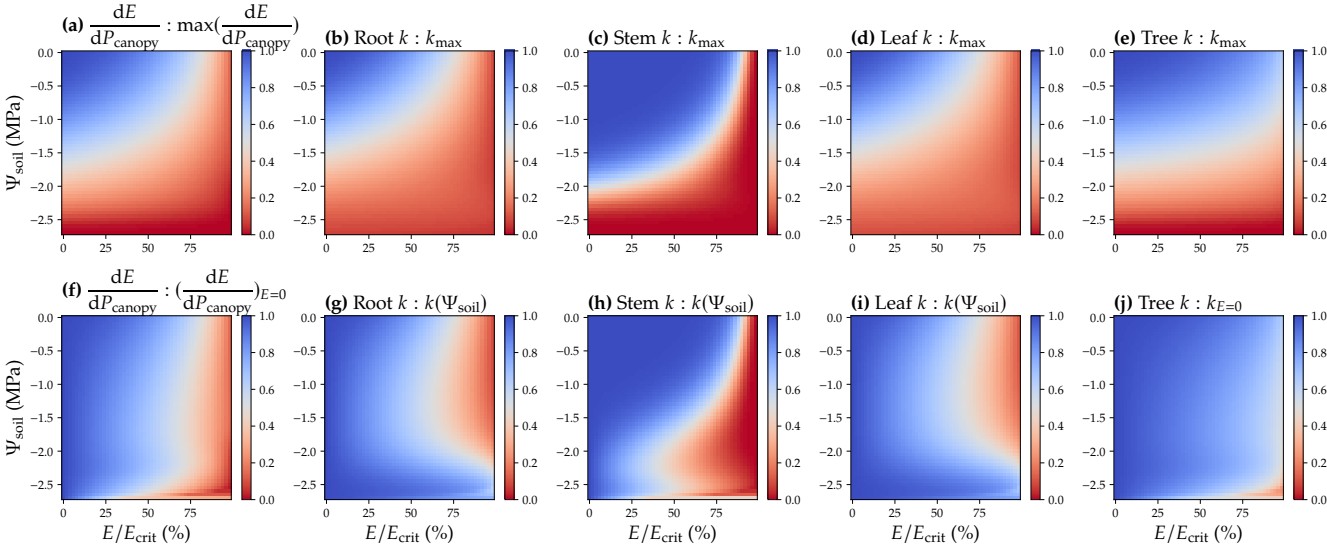

**Figure 3.** Comparison of different risk measures of stomatal opening. **(a)–(e)** Values are relative to the maximum when soil water potential is 0 and transpiration rate is 0. **(f)–(j)** Values are relative to the maximal when transpiration rate is 0 at the given soil water potential. The simulation is done assuming there is no drought legacy effect from previous xylem embolism. For the simulation, the plant has a root:stem:leaf resistance ratio of 2:1:1; root and stem height change are 1 and 10 m, respectively; VCs are represented using a Weibull function; Weibull B in MPa and C are 1.879 and 2.396 for root, 2.238 and 9.380 for stem, and 1.897 and 2.203 for leaf (data from Wang et al., 2019). See Table 1 for the list of symbols.

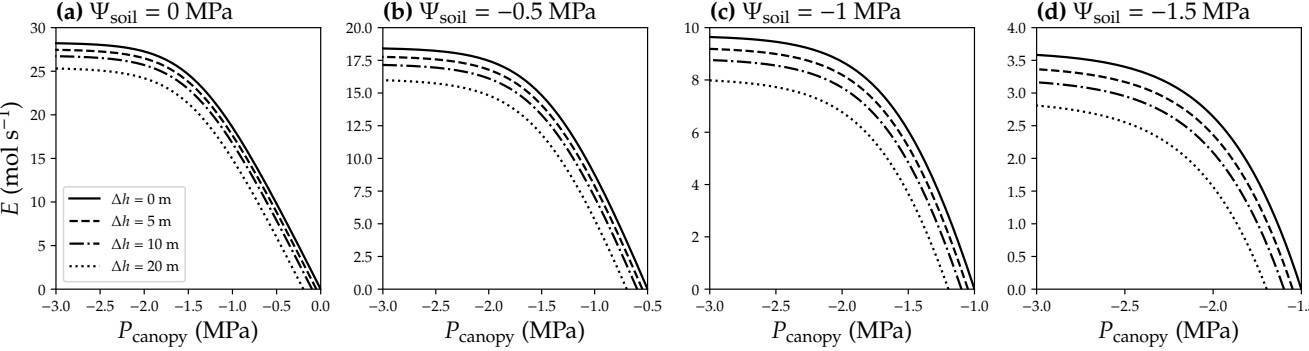

**Figure 4.** Water supply curve at different $\Psi_{\text{soil}}$ for xylem with different height change. The simulation is done assuming there is no drought legacy effect from previous xylem embolism. See Table 1 for the list of symbols.

## 8 Whole-plant vulnerability "curve"

The ideas of whole-plant conductance and VC largely advance the understanding of how plant traits coordinate as they provide a simple way to correlate different traits (see McCulloh et al. (2019) for an overview). Whole-plant hydraulic conductance ($k_{\text{plant}}$) depends on not only the upstream water potential (namely $\Psi_{\text{soil}}$) but also the downstream water pressure ($P_{\text{canopy}}$):

$k_{\text{plant}} = f(\Psi_{\text{soil}}, P_{\text{canopy}}) = \dfrac{E}{\text{DP}}$. However, one should be aware of the hidden assumptions when using the term whole-plant hydraulic conductance (or any similar terms): upstream water potential (soil water potential in this scenario) is the same everywhere, and driving pressure is the same everywhere, regardless of plant height, canopy light conditions, and root/stem/leaf network. Therefore, in the practice of modeling or research, the two assumptions are barely met. Further, note that $k_{\text{plant}}$ is a extensive parameter from root to leaves, and xylem water pressure and xylem hydraulic conductance are profiles rather

than being constant along the flow path. Therefore, by definition, there is not a whole-plant vulnerability curve; instead, $f(\Psi_{\text{soil}}, P_{\text{canopy}})$ is a whole-plant vulnerability surface (Fig. 5). It is obvious that none of $\Psi_{\text{soil}}$, $P_{\text{canopy}}$, or a mean pressure can predict a unique $k_{\text{plant}}$ (although the change of $k_{\text{plant}}$ is relatively smaller for the mean pressure; dotted line in Fig. 5). Further, drought legacy effect from previous non-refillable xylem embolism (Anderegg et al., 2015) would further complicate the scenario as the "surface" changes with drought legacy. Therefore, it is not recommended to use the term whole-plant

vulnerability curve in research.

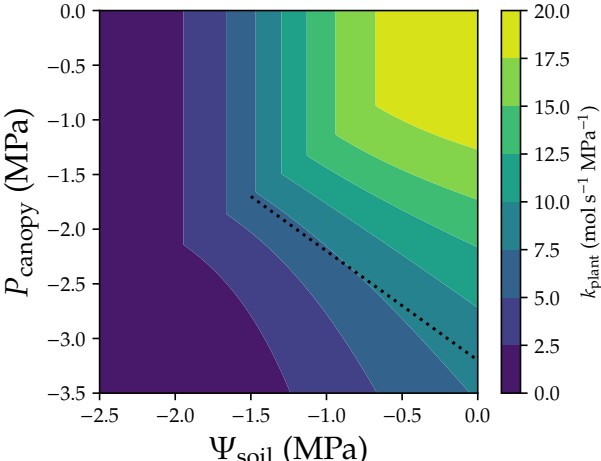

**Figure 5.** Whole-plant vulnerability surface. Whole-plant hydraulic conductance is computed using $k_{\text{plant}} = \dfrac{E}{\Psi_{\text{soil}} - P_{\text{canopy}} - \rho g h}$. The dotted line plots the scenario when mean xylem water pressure is $-1.6$ MPa. The simulation is done assuming there is no drought legacy effect from previous xylem embolism. See Table 1 for the list of symbols.

## 9 Temperature effects

When modeling plant hydraulics, the temperature effects on viscosity ($\eta$), surface tension ($\gamma$), and osmotic potential are typically ignored. However, when water temperature decreases from 25 °C to 10 °C (298.15 K to 283.15 K), (1) viscosity of water increases by 43.8%, meaning a $> 40\%$ increase in pressure drop along the flow path for a given flow rate; (2) surface tension of water increases by 3.1%, meaning that capillary force withholding the air-water interface at the pit membrane increases by 3.1% for a given curvature radius (xylem becomes more resistant to cavitation) and that soil metric potential becomes 3.1% more negative for a given soil water content; (3) soil osmotic water potential would be 5.0% less negative for a given ion concentration. Therefore, a more reasonable way to describe a xylem VC (e.g., using Weibull function) should be:

$$k = k_{\mathrm{max},25} \cdot \frac{\eta_{25}}{\eta} \cdot \exp\left[ -\left( \frac{-P}{B_{25}} \cdot \frac{\gamma_{25}}{\gamma} \right)^{C_{25}} \right], \tag{14}$$

where the subscript $_{25}$ denotes the values are at a reference temperature of 25 °C. In other words, $k_{\mathrm{max}}$ needs to be scaled to $k_{\mathrm{max},25} \cdot \eta_{25}/\eta$, and $P$ needs to be scaled to $P \cdot \gamma_{25}/\gamma$.

## 10 Conclusions

Plant hydraulics is often improperly represented in research, potentially resulting in ambiguities to those who are not familiar with the terminologies. This paper documents differences among commonly seen ambiguous and miscellaneous terms that are often not recognized, and the mistakes and misunderstandings researchers may make when using established methods and models. The mathematics and visualizations of the documented items will help researchers in their research and teaching associated with plant hydraulics.

*Author contributions.* All authors have contributed equally to the manuscript.

*Competing interests.* No competing interests

*Acknowledgements.* We gratefully acknowledge the generous support of Eric and Wendy Schmidt (by recommendation of the Schmidt Futures) and the Heising-Simons Foundation. This research has been supported by the National Aeronautics and Space Administration (NASA) Carbon Cycle Science grant 80NSSC21K1712 awarded to Christian Frankenberg. We benefit from the discussion with Dr. Tom Buckley for the proper usages of water potential and water pressure.

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
