# Peer review of "Technical note: Common ambiguities in plant hydraulics"

_Biogeosciences, 2022_

## Author Response (AR1)

**RC1**

The technical note from Yujie Wang and Christian Frankenberg focuses on still very poorly investigated area of modern plant ecology and hydrology focusing on describing and parameterizing the plant hydraulic properties as key parameters for simulation of plant or canopy transpiration and/or water uptake.

The paper is well written and can be interesting for modelers of plant hydrology to parameterize the transpiration and water transport in plant communities. The manuscript is in scope of J. Biogeoscience and can be publish in the journal after some revision.

[RESPONSE]
We thank reviewer 1 for the recommendation, and we have revised the manuscript carefully to add more items where researchers need to be cautious. See our detailed response below.

Actually, I guess a few points have to be additionally discussed in the paper.

1. All tall plant and trees are characterized by a non steady-state water transport through the soil - root- stem -branch - atmosphere system, i.e. the water fluxes at different plant segments is different e.g. root water uptake is not equal transpiration at some short time intervals. Plant tissue and leaves accumulate water which can later be used for transpiration…

[RESPONSE]
Modeling plant hydraulics in steady state or non-steady state is an option for users. In fact, modeling non-steady state flow will be more realistic. However, this requires more complicated models as the flow rate is not consistent any more. Further, the model parameterization is much more difficult. Researchers should choose the model with adequate complexity, otherwise the results may not be reliable. We mention this in a new section in the revision. Further, to better illustrate what we meant, we cross-ref the section to Figure 3 (section 6 in revision). Changes related (main text may differ slightly if we receive new comments from the reviewers and community, hereafter):

- "Plant hydraulic models have various complexities depending on the various aims of research and difficulties in model parameterization (Tyree and Ewers, 1991; Tyree and Zimmermann, 2002). In terms of flow profiles, the models can be categorized to steady state and non-steady state models. The steady state models use a constant flow rate within roots, stem, and leaves. The non-steady state models employ a changing flow rate within or among different tissues given the water exchange between xylem and capacitance tissues. In terms of the model complexity, the models range from a single element to a xylem network (say multiple roots and multiple canopy layers). Further, hydraulic conductance of an element may change with the growth of plants; for example, the drought legacy, maximum hydraulic conductance, and VC vary with the stack of new tree rings (McCulloh and Sperry, 2005; Cai and Tyree, 2010). Although more complex models may better represent the water flow and pressure profiles within the plants, increasing difficulties in model parameterization makes these more complex models less appealing to users. However, inappropriate model selection

could result in biased results, for instance, modeling plant hydraulics at steady state for plants with high water capacity and ignoring vessel tapering effect when modeling xylem growth. Thus, it is important to select plant hydraulic models with adequate complexity in topical research. See the section below for a detailed example of how reduced model complexity (ignoring VC segmentation) may bias the modeled hydraulic risk and thus stomatal responses."

2. Xylems of woody plants are very heterogeneous and characterized by different hydraulic conductance (for example along radial profile). Ignoring such effect can result in biased model results.

[RESPONSE]

We have this issue addressed along with the one above, as they are part of model complexity. Changes related:

- "In terms of the model complexity, the models range from a single element to a xylem network (say multiple roots and multiple canopy layers). Further, hydraulic conductance of an element may change with the growth of plants; for example, the drought legacy, maximum hydraulic conductance, and VC vary with the stack of new tree rings (McCulloh and Sperry, 2005; Cai and Tyree, 2010)."

3. One of a key objectives of your study is to "highlight the commonly seen ambiguities and/or misunderstandings in plant hydraulics" including different sections and particularly the "(4) stomatal model representations". Unfortunately this section is very poorly discussed in the manuscript.

[RESPONSE]

Thanks for pointing this out, and we have now renamed section 5 to "Stomatal model representation", and added more description. Changes related:

- Section "Stomatal model representation".
- "Plant hydraulics-based stomatal models are gaining increasing interest in the vegetation and land modeling communities (e.g., Kennedy et al., 2019; Sabot et al., 2020) as they predict stomatal closure at dry environmental conditions without employing an arbitrary tuning factor (often known as the β factor) (Powell et al., 2013). For instance, the recently developed optimality theory-based models propose that plants should balance the gain and risk associated with stomatal functioning (Wolf et al., 2016; Sperry et al., 2017). When plants open their stomata more, plants gain more photosynthetic carbon, but lose more water and have higher risk in hydraulic failure; therefore, plants are supposed to find a sweet zone to maximize the difference between the gain and risk. These optimality theory models, particularly those weigh the risk based on plant hydraulics, show comparable or better predictive skills compared to the statistical approaches (Anderegg et al., 2018; Eller et al., 2018; Venturas et al., 2018; Wang et al., 2020; Sabot et al., 2022). However, a common mistake when using plant hydraulics-based models is that one does not follow the original model formula or hypothesis."

Specific comments.

"The risk of stomatal opening" is not the best term for the sentence from ecological point of view. Stomatal opening and closing are very important physiological processes in plants.  It is better to use the term e.g. "stomatal response", "stomatal functioning", or any.  So, I suggest to reformulate the sentence.
[RESPONSE]
We have added more description related to stomatal optimality theory, and reworded this sentence as well to be more clear. Changes related

- "For instance, the recently developed optimality theory-based models propose that plants should balance the gain and risk associated with stomatal functioning (Wolf et al., 2016; Sperry et al., 2017). When plants open their stomata more, plants gain more photosynthetic carbon, but lose more water and have higher risk in hydraulic failure; therefore, plants are supposed to find a sweet zone to maximize the difference between the gain and risk."
- "For example, the Sperry et al. (2017) model defines the risk associated with stomatal functioning ($\Theta$) as"

**CC1**

This manuscript propagates some common misconceptions about plant water relations, concerning the roles of diffusion vs advection (and water potential vs pressure) in xylem water transport, and the meaning and relevance of the gravitational potential term in the formal definition of water potential.

1. line 39-40: "Water potential gradients drive water flow through permeable media such as xylem conduits"

This is incorrect, except on extremely small scales. Water potential describes the tendency for water to move between adjacent phases (regions of internally uniform thermodynamic states) due to the net, or average, movement of individual water molecules. In other words, it predicts where water molecules will diffuse. It does not describe the movement of coherent bodies of water under the action of body forces (advection). Water potential gradients thus drive liquid water movement only at spatial scales where diffusion is faster than advection. Those scales are extremely small, and can be quantified using the Peclet number, which is the ratio of advective to diffusive transport velocities. The Peclet number equals V*X/D, where V is the advective velocity, X is the distance and D is the molecular diffusivity (of water in liquid water, in this case; about 2.4e-9 m2 s-1). If the Peclet number is greater than 1, advection dominates; if it is less than 1, diffusion dominates. For example, consider water moving in the xylem at a velocity of 1 cm/hr (2.8e-6 m/s), which is quite low but not negligible. In this case the Peclet number is unity for X around 0.9 mm; that is, for distances over a millimeter, advection dominates and thus pressure gradients are the more relevant driver of water movement. (For more typical midday sap velocities of ~10-50 cm/hr, the Peclet number is unity for X around 20-90 microns.) Thus, advection dominates xylem water transport in nearly all cases, so the statement quoted above is precisely incorrect.
[RESPONSE]
Thanks for pointing out the mistake we made in the definition. The mistake was due to the fact water potential has been used in place of P+$\rho$gh for water mass flow. We agree with you that the use of "water potential" is wrong for mass flow. We also appreciate the

discussion via zoom. Here, we revise this part accordingly to clarify that (1) water pressure should be used for water mass flow and xylem cavitation, and (2) water potential should be used for diffusion. Changes related (main text may differ slightly if we receive new comments from the reviewers and community, hereafter):

- "Water movement in plants includes mass flow through xylem conduits and diffusion between xylem conduits and capacitance tissues. Water mass flow (from site 1 to 2) in xylem is driven by the net force at the target plane per area (driving pressure, DP), which is $DP = P_{x1} - P_{x2} + \rho g h_1 - \rho g h_2$ as in Figure 1a; and the dissolved ions only play a role in liquid water density. Water diffusion across the cell membrane (from xylem conduit to the cell) is driven by the potential difference, which is $P_x - P_c + \Psi_{mx} - \Psi_{mc}$ as in Figure 1a; and the dissolved ions play a role through the osmotic potential."

[Figure]

-
- "As ion concentration in xylem conduit is very low, the osmotic potential in xylem conduit ($\Psi_{mx}$) is often ignored. Further, as there is not a general name for the term $P_x + \rho g h$, water potential is imperceptibly used in place of mass flow driving pressure (i.e., DP) because of the gravity term in it. To date, many people use water potential rather than water pressure when modeling and describing mass water flow. This, though may be easier for people to understand, is not correct. For example, in Figure 1b, if the osmotic potential at the bottom and top are the same, the driving pressure and water potential difference are the same. However, if the osmotic potential at the top is more negative than at the bottom (for example, via adding a very thin layer of high concentration salt solution), the driving pressure will be lower than the water potential difference. Thus, using potential difference for water mass flow is technically incorrect, and it is necessary to clarify the terminology to distinguish them."

1. lines 44-46: "For instance, for a 100 m tall tree with no transpiration, leaf water potential is equal to soil water potential; however, leaf xylem water pressure would be approximately 1 MPa more negative than xylem pressure at the tree base;"

It is nominally correct, but misleading and pointless, to say that leaf water potential would be equal to soil water potential in this example. While it is true that chemical potential (and water potential, by its conventional definition) contains a gravitational potential term, that term is never relevant to plants on Earth. This is because gravitational potential never varies substantially at the spatial scales where water potential drives water movement. (The reason gravity does matter for water transport is addressed in my next comment, #3.)

We measure water potential because it tells us something about the physiologically relevant condition of water in a given tissue; and in the case of leaves, because it gives us an estimate of the pressure in the xylem water at that location – which is useful for predicting both long-distance water transport and xylem embolism. Any measurement of water potential (including by psychrometry) would give you a value of -1 MPa for the leaf in this example. The fact that this leaf's water had more potential energy than that of a leaf near the ground would not do the leaf any good, in terms of dealing with the negative consequences of its actual water status.

If we want to be at once rigorous, practical, and clear with our definitions, we should probably redefine water potential to exclude the influence of fields (like gravity) that never vary, in practice, at the spatial scales where water potential is a relevant driver of water movement. (There's nothing stopping us from redefining it. It was an arbitrary definition to begin with.)

[RESPONSE]

Thanks for encouraging us to redefine the terms, and it indeed makes things clear, as it is not correct to use pressure difference or potential difference to describe the water mass flow driving pressure. To avoid overly distracting readers, we replace this example with a new Figure 1 (pasted above) and a few suggestions. Changes related:

- "Water pressure difference between xylem water and surrounding air is responsible for air-seeded conduit cavitation (Sperry and Tyree, 1988; Tyree and Sperry, 1989), which occurs when the pressure difference exceeds the capillary pressure at the air-water interface. Thus, using water potential to describe xylem vulnerability curve should be avoided. Although the Δ values of the two are often interchangeable in many scenarios (e.g., when there is no height change or external air pressure), one needs to be cautious to avoid ambiguity: Use pressure in xylem cavitation; Use pressure in water mass flow; Use potential in water diffusion across the cell membrane (e.g., water exchange between xylem and living cell)."

1. Continuing from the above "...and using pressure drop here to derive flow rate will be incorrect when there is height change"

This is a red herring. Nobody uses pressure drop alone to derive flow rate in such a case (or if they do, it's rare - I've never seen it done in cases where the height change is very large). They properly subtract the gravitational head (the force per unit area caused by the weight of the water column). A simple force balance analysis for the xylem water is what leads to the actual flow equation (with flow being proportional to the pressure difference minus the gravitational head).

[RESPONSE]

In model practice, the gravity term is actually often neglected, because including the term makes it way more difficult to model plant hydraulics analytically. For example, transpiration rate through the xylem at a given pressure/potential gradient is often written as $E = \int k(P)\, dP$ or $E = \int k(Psi)\, dPsi$. In this case, it is incorrect to use either potential (because k is not a function of Psi) or use P (because of the gravity term). As a result, in many scenarios, we ignore the gravity term in the Sperry et al. (2017) and Venturas et al. (2018) studies. As the main points are to clarify where potential and pressure should be used, we replace the sentences with our suggestions. Changes related:

- "Although the $\Delta$ values of the two are often interchangeable in many scenarios (e.g., when there is no height change or external air pressure), one needs to be cautious to avoid ambiguity: Use pressure in xylem cavitation; Use pressure in water mass flow; Use potential in water diffusion across the cell membrane (e.g., water exchange between xylem and living cell)."

1. lines 51-53: "Leaf water potential is often estimated using the pressure chamber method (Scholander et al., 1964; Boyer, 1967). However, the term "potential" is not accurate here, as the pressure chamber method gives the applied pressure at the free meniscus of the cut end."

This is sophistic. Plant physiologists all understand that the pressure bomb gives the pressure in the xylem water (or, we hope, a decent estimate of it). They refer to it as "water potential" because (a) the xylem water generally has very low solute content, so its pressure is approximately equal to its water potential (ignoring the pointless gravitational term as discussed above), and (b) in the equilibrated leaf, the living cells' water potentials will be equal to that of the xylem water. Again, in this case, bringing up the gravitational component of water potential adds more confusion than clarity and is meaningless in practice.

[RESPONSE]

To avoid making the question overly complicated, we simply give our recommendations in the revision. Changes related:

- "A commonly seen sophistic "mistake" is the use of leaf water potential. While it is well known that the pressure chamber method (Scholander et al., 1964; Boyer, 1967) gives a decent estimate of the xylem water pressure, people often refer to the measurement as leaf water potential as (a) xylem conduit water has very low solute content, and (b) gravity term is often negligible compared to the very negative leaf xylem water pressure. However, it is always more accurate to treat it as an equivalent pressure or a balance pressure (at the end of xylem). For example, when the whole plant is under equilibrium, leaf water potential should be equal everywhere, but the measured leaf xylem pressure would differ for leaves at different height. Similar logic applies to xylem water potential and xylem water pressure, and so does the thermocouple psychrometers method (Boyer and Knipling, 1965; Boyer, 1968). It is recommended to refer to the measurement as leaf/xylem water pressure or balance pressure in the future, rather than leaf/xylem water potential."

1. lines 104-105 and Equation 11. "A common mistake when using plant hydraulics-based models is that one does not follow the original model formula or hypothesis"

Ironically, Equation 11 is not in fact the Sperry et al (2017) model. It defined Theta as Theta = (kcmax – kc(Pc))/(kcmax – kcrit).

[RESPONSE]
Thanks for pointing this out. We have now added a description why ignored the kcrit term in the equation. Changes related:

- "For example, the Sperry et al. (2017) model defines the risk associated with stomatal functioning (Θ) as"

$$\Theta = A_{max} \cdot \frac{k_{canopy,ref} - k_{canopy}(P_{canopy})}{k_{canopy,ref} - k_{crit}},$$

-
- "where Amax is the maximal achievable photosynthetic rate at the given setting, Pcanopy is the leaf xylem end water pressure in the canopy, kcanopy,ref is the maximum kcanopy when transpiration rate is 0, and kcrit is the kcanopy when transpiration rate is maximum. The kcrit by definition is 0, as a minimum incremental transpiration rate results in infinity increase in xylem pressure (the dE/dP = 0 in Figure 1). Thus, in the subsequent research where Sperry et al. (2017) was tested (Venturas et al., 2018; Wang et al., 2020), the model has been reformulated to"

$$\Theta = A_{max} \cdot \left[ 1 - \frac{k_{canopy}(P_{canopy})}{k_{canopy,ref}} \right].$$

- New equation 13     (old equation 11)

**EC1**

Dear Authors,
now that you received two comments to the manuscript, please use the opportunity to start discussing them already now. This may then lead to a fruitful interactive discussion.
Best wishes,
Andreas

[RESPONSE]
Thanks for the reminder. We are drafting our initial response, and will have it posted some time this week. However, the final manuscript may differ slightly if we receive new comments from the reviewer or community.

---

## Author Response (AR2)

**Reviewer #1**

The paper was significantly revised and improved.
There are a few technical comments to the chapter 6, only.

Line 150
I guess the term "Θ - the risk associated with stomatal functioning" can be replaced by the term used in Sperry et al 2017 - transpirational cost function.
[Response]
In Sperry et al. (2017), we called it cost. However, since the term is more a shadow cost that does not show up in real instantaneous carbon gain (as pointed out in Buckley (2017) https://doi.org/10.1104/pp.16.01772), we started to use the term risk as used in Venturas et al. (2018).

Line 154
The term "the leaf xylem end water pressure" needs some explanation.
[Response]
We clarified this in lines 150-151 (clean revision, hereafter) that "$P_{canopy}$ is the water pressure at the end of leaf xylem" (also in our new Table 1).

Line 156
"transpiration rate is maximum"
May be - expressions 'all stomata are completely open' or 'maximum stomatal conductance' - are more correct?
[Response]
Stomatal conductance may not be able to fully open during drought stress. For example, if maximal stomatal conductance is 0.3 mol $m^{-2}$ $s^{-1}$, xylem may not even cavitate at well watered condition. In another case, xylem may lose all its conductivity even when stomata are not fully open during a drought. Therefore, we are using the physical limitation to xylem to describe kcrit. We clarified this in lines 151-152 that "$k_{crit}$ is the $k_{canopy}$ when transpiration rate reaches the maximum transport capacity of the xylem".

**Reviewer #2**

32: ...focuses... ... is not encouraged...
[Response]
Thanks. We have made the change in the revision.

37-38: It also includes diffusion among mesophyll cells in the liquid phase, and diffusion through the mesophyll in the vapor phase. Vapor-phase transport affects water potentials of all tissues downstream from the leaf xylem, so it must be considered part of "water transport" even though we may prefer to think of "water transport" as ending at the sites of evaporation. Water transport in plants may also include liquid-phase bulk flow through the apoplastic spaces in the mesophyll, and either diffusion or effusion through plasmodesmata, although the importance of those latter two pathways is largely speculative.
[Response]
Thanks for correcting this. We revised the sentence to "Water movement **in xylem conduits** includes mass flow through xylem conduits and diffusion between xylem conduits and capacitance tissues" to focus on xylem water movement only (lines 37-38, clean revision, hereafter). We also added a new paragraph to highlight that the water potential gradient should be used for diffusion among living cells and that water pressure difference should be used for mass flow in the apoplastic spaces: "We note that water transport in plants also include mass flow within the apoplastic spaces (e.g., in roots and leaves; Aloni et al. (1998)) and through plasmodesmata (e.g., between bundle sheath and phloem; Schulz (2015)), liquid water diffusion among living cells, and gaseous vapor phase diffusion among water-air interfaces (e.g., vapor diffusion within the stomatal chamber; Buckley (2015); Buckley et al. (2017)). As recommended, it is more accurate to use potential for diffusion and pressure to mass flow" as in lines 66-70.

39: please define the symbols in the text here, or better, make a table listing all symbols, units, etc., and then refer to the table at line 39.
[Response]
Thanks for the suggestions, and we added a new Table 1 (pasted below).

| Symbol | Description | Unit |
| --- | --- | --- |
| DP | Driving pressure ($P_{x1} - P_{x2} + \rho g h_1 - \rho g h_2$) | MPa |
| $P$ or $p$ | Water pressure | MPa |
| $P_c$ | Cell turgor pressure | MPa |
| $P_x$ | Xylem water pressure | MPa |
| $\Psi$ or $\psi$ | Water potential | MPa |
| $\Psi_s$ | Osmotic potential from dissolved solute | MPa |
| $\Psi_{sc}$ | $\Psi_s$ of living cells | MPa |
| $\Psi_{sx}$ | $\Psi_s$ of xylem sap | MPa |
| $A_L$ | Leaf area the xylem supports | $m^2$ |
| $A_S$ | Sapwood area | $m^2$ |
| $k$ | Hydraulic conductance | $mol\ MPa^{-1}\ s^{-1}$ |
| $k_{branch}$ | Hydraulic conductance of the branch | $mol\ MPa^{-1}\ s^{-1}$ |
| $k_L$ | Leaf area specific hydraulic conductance | $mol\ MPa^{-1}\ m^{-2}\ s^{-1}$ |
| $K$ | Hydraulic conductivity | $mol\ m\ MPa^{-1}\ s^{-1}$ |
| $K_L$ | Leaf area specific hydraulic conductivity | $mol\ MPa^{-1}\ m^{-1}\ s^{-1}$ |
| $K_S$ | Sapwood area specific hydraulic conductivity | $mol\ MPa^{-1}\ m^{-1}\ s^{-1}$ |
| $Q$ | Flow rate through the xylem segment | $mol\ s^{-1}$ |
| $A_{max}$ | Maximum achievable photosynthetic rate | $\mu mol\ m^{-2}\ s^{-1}$ |
| $E$ | Transpiration rate of the whole plant | $mol\ s^{-1}$ |
| $E_{crit}$ | Maximum $E$ beyond which the plant desiccate | $mol\ s^{-1}$ |
| $k_{canopy}$ | Marginal hydraulic conductance of the canopy ($dE/dP_{canopy}$) | $mol\ MPa^{-1}\ s^{-1}$ |
| $k_{canopy,ref}$ | $k_{canopy}$ when transpiration rate is 0 | $mol\ MPa^{-1}\ s^{-1}$ |
| $k_{plant}$ | Whole plant hydraulic conductance | $mol\ MPa^{-1}\ s^{-1}$ |
| $P_{canopy}$ | Water pressure at the end of leaf xylem | MPa |
| $\Psi_{soil}$ | Soil water potential | MPa |
| $\Theta$ | Risk associated with stomatal opening | $\mu mol\ m^{-2}\ s^{-1}$ |
| VC | Vulnerability curve | - |
| $a, b$ | Logistic function parameters | -, $MPa^{-1}$ |
| $B, C$ | Weibull function parameters | MPa, - |
| $k_{max}$ | Maximum hydraulic conductance | $mol\ MPa^{-1}\ s^{-1}$ |
| $k_{max,25}$ | Maximum hydraulic conductance at 25 °C | $mol\ MPa^{-1}\ s^{-1}$ |
| $m, n$ | Power function parameters | $MPa^{-n}$, - |
| $P_{50}$ | Xylem water pressure where xylem loses 50% conductance | MPa |
| $\eta, \eta_{25}$ | Viscosity of water (at 25 °C) | Pa s |
| $\gamma, \gamma_{25}$ | Surface tension of water (at 25 °C) | $N\ m^{-1}$ |

39: at the end of this sentence, it may help to explain for readers why you're saying this about the ions: namely, that osmotic potential does not contribute to the driving forces for long-distance transport in the xylem, because (1) the reflection coefficient in the xylem is zero, and (2) the distances involved are too long for diffusion and thus chemical potential to be an important contributor.

[Response]

Reason (1) for the reflection coefficient would require a too detailed (and distracting) explanation to most readers, so we preferred not to include it in the revision. However, see our revised text "Dissolved ions only play a role in liquid water density but do not contribute to the driving pressure for long-distance transport in the xylem, because the distance involved is too long for diffusion and thus chemical potential to be an important contributor." as in lines 39-41.

41: The symbol "psi_m" for osmotic potential is likely to generate confusion. I suggest you use "psi_s" or reverse the sign convention and use greek "pi" for osmotic pressure. psi_m looks like matric potential or mesophyll water potential to me.

[Response]

We changed the symbol from Psi_m to Psi_s (text and figure).

[Figure]

line 43: There is actually a general name for Px + rho*gh, which is the "head" (or hydraulic head).

[Response]

We thought about using water head that has a similar definition in our last revision, but they differ in their units. We removed the sentence and revised the sentence to "Therefore, water potential is imperceptibly used in place of mass flow driving pressure (i.e., DP) because of the gravity term in it." as in lines 44-45.

48-49: This example illustrates *how* delta psi might not equal delta P, but it doesn't explain *why* delta P is the correct driving force in this context instead of delta psi. That explanation is

the one I gave in my first review, about the relevance of different modes of transport (bulk flow vs diffusion) at different spatial scales. Since the purpose of this ms is to clarify these issues and prevent confusion and error, I really think it's necessary to explain this point explicitly.
[Response]
We added the sentence "Besides the fact that the values of DP and water potential difference do not always equal, the primary reason for not misuse them is that water potential describes the tendency for water to move between adjacent phases (where water molecules will diffuse), whereas pressure is more relevant to bulk water movement" as in lines 50-53 to clarify that the reason to not use potential in place of pressure is that they are responsible for diffusion and bulk flow, respectively.

56: The criteria for the delta values to be equal are a bit more strict than this – it also assumes the osmotic potential in the xylem is zero. It's typically pretty small, as you noted earlier, but not zero. And there are circumstances where it's not negligible.
[Response]
We made the criteria more strict in the revision: "(e.g., when there is no height change or external air pressure, and osmotic potential in the xylem is zero)" as in lines 61-62.

61: I used the word "sophistic" in my review in a different context; I wouldn't say the use of leaf water potential is a "sophistic mistake". You could say a "mistake of interpretation" or just mistake.
[Response]
We revised the sentences to "A commonly seen mistake is the use of leaf water potential to describe measurements from the pressure chamber method (Scholander et al., 1964; Boyer, 1967), which gives a decent estimate of xylem water pressure. People often refer to the measurement as leaf water potential as (a) xylem conduit water has very low solute content, (b) gravity term is often negligible compared to the very negative leaf xylem water pressure, and (c) if the water has reached equilibrium internally prior to the pressure chamber measurement, xylem water potential should equal that in the mesophyll" as in lines 71-75.

Also, on 61, the first sentence needs to be linked to the second to make sense. Perhaps "A commonly seen mistake is the use of leaf water potential to describe measurements from the pressure chamber method (Scholander...), which gives a decent estimate of the xylem water potential. People often refer to this measurement as leaf water potential because..."
[Response]
See our response above.

63: should add to this list: (c) provided the leaf has been allowed to equilibrate internally prior to the pressure chamber measurement, the water potential in the xylem should equal that in the mesophyll.
[Response]
See our response above.

65: "For example, when the whole plant is under equilibrium, leaf water potential should be equal everywhere, but the measured leaf xylem pressure would differ for leaves at different height." The first part of this sentence is incorrect (if you interpret water potential as a measurable) or inadvertently misleading (if you interpret it to include the un-measurable "gravitational component"). *Measured* leaf water potential (e.g. by psychrometry or the Shardakov method) would not be equal everywhere; assuming no solutes in the xylem, leaf water potential would necessarily equal xylem water potential everywhere, which in turn would equal xylem pressure (if no solutes). Now, that's not true if you include the 'gravitational component', but including that term is always confusing because *it cannot be measured* – any method of measuring WP involves chemically equilibrating the leaf's water with some other pool of water whose potential is either known (as in Shardakov) or can be measured (as in the vapor in the psychrometer chamber), and that act of chemical equilibration requires the two pools of water to be close enough that the gravitational component can't differ measurably. Something that cannot be measured in practice and only exists in a theoretical definition is only going to generate confusion... unless you tackle the confusion head-on. I realize this issue is confusing for readers, but again, if this ms is to serve its intended purpose, it should tackle these difficult issues openly!
[Response]
The definition of water potential is the sum of water pressure, osmotic potential, and gravity component. However, the measurements do not include the gravity component, and a "true" water potential has to be computed. The main idea of the sentence is that we should refer to the pressure chamber measurements as estimates of xylem pressure. As the idea has been conveyed already, we removed the sentence to avoid confusion.
We clarified this in the revision: "It is recommended to refer to the measurement as leaf/xylem water pressure or balance pressure in the future, rather than leaf/xylem water potential that is not directly measurable." as in lines 77-79.

73: I think the proper term here is "extensive" properties, not bulk properties. They are extensive because their numerical values depend on the extent (size) of the system. e.g., flow rate will be greater for 10 xylem conduits taken together than for just one conduit, whereas flux (flow per cross sectional area) would be the same in both cases, and would be an "intensive" quantity.
[Response]
Thanks for the suggestion. We revised the text to "Hydraulic conductance (flow rate divided by driving pressure) is an extensive property (depends on the extent/size of the system), whereas hydraulic conductivity is an intensive property that is supposed to represent different xylem anatomy" as in lines 82-84.

74: It's true that xylem structure is often highly non-uniform, but you could say the same thing about electrical properties of most realistic things in nature. The electrical vs hydraulic distinction doesn't really clarify here. What matters is that "conductivity" is meant to describe an intensive property.
[Response]
See our response above.

76: Using "E" for flow rate here could confuse readers, because it so often means "transpiration rate", which in turn is conventionally expressed on a leaf area basis – so k would then look like a conductivity because E would look like a flux.

[Response]

Thanks for pointing it out. I have the tendency to define E as flow rate, and $E_{leaf}$ as transpiration rate per leaf area. To not confuse readers, we used Q for the hydraulic conductance equations, and E as the whole plant transpiration in the paper (Table 1).

77: I'm not sure what you mean by "and area not accounted for by k".

[Response]

We clarified this in the revision: "(a) hydraulic conductance (namely k) is the ratio between flow rate through the segment (Q) and driving pressure ($\Delta P - \rho g \Delta h$) (an extensive parameter depending on segment length and cross-section area), (b) hydraulic conductivity (namely K) is the ratio between flow rate and driving pressure gradient (an extensive parameter depending on segment cross-section area)" as in lines 84-87.